# We Are Concerned about the Future and We Are Here to Support the Change: Let’s Talk and Work Together!

**DOI:** 10.3390/children9101574

**Published:** 2022-10-18

**Authors:** Cátia Branquinho, Luís Lobo Xavier, Catarina Andrade, Tomás Ferreira, Catarina Noronha, Tony Wainwright, Margarida Gaspar de Matos

**Affiliations:** 1ISAMB (Environmental Health Institute), Aventura Social, University of Lisbon, 1649-026 Lisbon, Portugal; 2#GenerationsWithAVoice Project, Aventura Social & Calouste Gulbenkian Foundation, Portugal; 3Calouste Gulbenkian Foundation, 1067-001 Lisbon, Portugal; 4Department of Psychology, University of Exeter, Exeter EX4 4QG, UK

**Keywords:** youth participatory programs, generational inequalities, intergenerational dialogue, intergenerational justice, future concerns, sustainability

## Abstract

Scarcely explored, intergenerational dialogue may support the re-encounter between generations. Background: Focused on intergenerational sharing and on the identification of differences between generations, the project #GenerationsWithAVoice aims to identify generational inequalities, with the aim of promoting awareness of intergenerational challenges, boosting public debate and interest in public policies. Methods: Twenty focus groups were developed, and an evaluation instrument was applied. Results: (i) Young people believe that they have more knowledge, but less propensity for action, leaving this task to the following generations; (ii) the family and housing emerge as the issues with the greatest number of problems identified, but also as important resources; (iii) government and politics, community and society, and the economy are of less interest and knowledge on the part of this generation; (iv) school seems to be the ideal scenario for the implementation of strategies that lead to change. Conclusions: We highlight the role of this work in the deconstruction of beliefs regarding previous generations, the development of knowledge, and the promotion of cohesion and social support.

## 1. Introduction

Although not widely studied, the concept of intergenerational justice, often referred to as intergenerational dialogue, intergenerational equality, intergenerational equity, intergenerational sustainability, or intergenerational solidarity, relates to the concept that the search for the well-being of today’s generations should not compromise the opportunities and resources of the following generations.

Comprising the distributive, procedural, restorative, and retributive dimensions [1], the concept of intergenerational justice can be applied to several areas that may impact human well-being [2,3,4], and can be conceptualized in four major areas: (i) basic human needs; (ii) economic needs; (iii) environmental needs; and (iv) subjective well-being [5]. According to these authors, a sustainable world requires that the human needs of current generations are met without compromising the possibility of future generations to meet their needs as well.

When we talk about intergenerational justice, we still talk mainly about areas such as public debt, social security, the labor market, and the environment [6,7], but it is imperative to include other areas with implications for present and future generations, such as family and housing, education, health, community and society, government and policies, and culture. Caney’s [8] view also follows this line, arguing that analyzing and intervening independently in these areas may be too difficult and have negative consequences, as policy responses to one of these areas may affect the others.

The incentive for intergenerational justice involves not only raising awareness and providing education to current generations, but also implementing public policies that protect future generations [9]. This is starting to be a priority for the OECD [10] in creating more and better opportunities for young people. According to Vanhuysse [7], the implementation of policy measures such as (i) tax and social benefits or subsidies to support parents or caregivers; (ii) taxes and regulation of carbon emissions; (iii) an investment that enables a better education of the next generations and the creation of educational programs; and (iv) the existence of a proxy vote that could be used by the parents or caregivers of the minor, could promote greater intergenerational justice. In complement, Ribeiro [11] adds the creation of a constitutional protection to protect the following generations, imposing constitutional limits in the current generations. Based on a recent systematic literature review, the solution to greater intergenerational justice lies in closing the generation gap from a social and historical perspective; raising awareness and caring for current generations; and encouraging intergenerational dialogue, cohesion, and co-action [12].

### 1.1. Youth Engagement and Participation

Seen as a fundamental right [13] enabling involvement in the development process itself, greater knowledge of human rights, and more active citizenship [14], youth participation has gained greater prominence in recent decades.

Some of the most well-known theories (e.g., Roger Hart’s [15] “Participation Ladder”, Phil Treseder’s [16] “Levels of Participation”, Harry Shier’s [17] “Participation Pathways”) have served as inspiration for integrating youth participation into research. 

The conceptualization of these programs, created by Rodriguez and Brown [18], determines three principles: (a) their focus is established on the life experiences and concerns of young people; (b) the methodological and pedagogical processes are based on the collaboration and participation of young people; and (c) they have a transformative character, contributing to change theories and practices that improve the lives of young people, along with their communities.

Based on the premise that this generation has valuable knowledge about their problems and needs [19,20,21,22] and that they should give their voice in the first person based on their experience [23], participatory research programs have proven to be effective interventions in the processes of individual development, health, and community. According to a systematic literature review conducted by Anyon et al. [24], these programs have important impacts on agency, leadership, academic (or professional), social, interpersonal, and cognitive levels.

In a contrasting model to traditional paradigms, in youth participatory research programs, the researcher no longer speaks for the young person [25], but the young person benefits from the adult–young person mentoring relationship, sharing power during the process, developing their skills and consultancy, encouraging debate based on social change and the establishment of alliances with stakeholders [26].

### 1.2. #GenerationsWithAVoice

Recognizing (i) the role of the Y–AP (youth–adult partnership) approach in conceptualizing problems and implementing actions [27,28]; (ii) that youth voice, choices, and participation are fundamental to the sustainability of the future [1]; and that (iii) intergenerational justice, i.e., the notion that the pursuit of well-being by current generations should not jeopardize opportunities for the well-being of subsequent generations, the #GenerationsWithAVoice project is presented, integrated in the Future Forum initiative of the Calouste Gulbenkian Foundation, which aims to investigate and provide knowledge about tomorrow’s challenges, anticipating their arrival and encouraging public debate.

Centered on a reciprocal intergenerational sharing (young–adult and adult–young), this project aims to identify generational inequalities, focused on areas such as the environment (including climate change and planet protection), family and housing, education, health, work, economy, community and society, government and politics, and culture, with the aim of promoting awareness of intergenerational challenges, boosting public debate and interest in public policies, thus contributing to a more positive, fair, equitable, and active society.

This project has had the following objectives: (i) to investigate the dominant opinions of the younger generations on the topics under study; (ii) to promote intergenerational dialogue, stimulating a sharing of knowledge between generations; (iii) to understand young people’s views on strategies to promote intergenerational justice, and on how to influence decision-making processes based on their concerns and interests; and (iv) to know the perspective of young people on how to influence institutional and public policies according to their concerns and interests.

In an attempt to apply this premise from an early age, the research team of this project is currently developing the Learn to Fly project, which aims to develop greater social participation and dialogue between generations through identification and action on relevant societal issues. In addition, it is anticipated that Learn to Fly will develop psychological flexibility, openness, curiosity, autonomy, and self-regulation in children at the beginning of their school career (5–6 years). With the inclusion of educators, families, and children, it is intended that, after evaluation, it will be included in the pre-school and school curriculum [29].

In this study, generation was considered as a group of people born in a given year or period.

## 2. Materials and Methods

### 2.1. Procedure and Participants

Developed during the period March 2020 to April 2021, during a pandemic scenario, #GenerationsWithAVoice saw all its methodology adapted to the online format.

Initiated with an awareness raising event, #GenerationsWithAVoice began by inviting 15 participants (young people, health and education professionals linked to youth work) for an interview focused on the themes under study and the role of intergenerational justice, in order to develop a storytelling video for use in dissemination of the project in academia, and for raising awareness in the community. The 15 interviews had an average duration of 32 min, resulting in the production of 5 videos: environment; family and housing; employment; economy; and digital generation.

The 15 participants were recruited through direct invitation via email, based on their experience and expertise, and the interviews were conducted in the period from May to October 2020.

Nine months later, having accepted that the face-to-face methodology would be impossible, contacts were reactivated with five schools (one for each territorial unit in mainland Portugal) contacted at the start of the project to facilitate a focus group with young people born in 2002 (voters) and another focus group with young people born in 2004 (non-voters), and focus groups were started online. The rationale for these age groups was understood to be the fact that becoming a “voter” would increase young people’s awareness regarding society, societal issues, and community engagement. 

The participating schools were suggested by the Directorate-General for Education, and the participating classes were selected by the director of the educational institution, based on the availability of a class teacher to support the groups, and the age of the students (born in 2022—12th grade and in 2004—10th grade). The focus groups were conducted from November 2020 to January 2021.

A total of 20 focus groups were conducted (two groups per region × two times, before and after the intergenerational dialogue), comprising a minimum number of 4 and a maximum number of 22 participants, bringing together young people aged 16–17 years old (groups corresponding to the youngest group), and 18–19 years old (older). The groups had an average duration of 58 min.

The groups were run at two different times for each of the groups in each school (16–17-year-olds and 18–19-year-olds), with an interval of two weeks. During this period, the challenge was to encourage intergenerational dialogue through interviews with parents, uncles, grandparents, teachers, or other people from the “adult” and “older” generations of their community, in order to recount their memories and experiences related to the themes under study.

This dialogue was intended to stimulate debate in the second focus group and to facilitate the identification of solutions to the problems and needs identified in the first focus group.

To support how intergenerational justice was understood by the participants, a questionnaire was developed to assess the importance, knowledge, concern, and action that the young participants attributed to each of the topics studied, along with their perception of future generations, their parents’ generation, and their grandparents’ generation. Although this questionnaire was aimed at a pre- (focus group 1) and post-test (focus group 2) study, due to the constraints within the COVID-19 pandemic, it was only possible to collect 32 reports from a single point in time (17.2 years; *SD* = 0.998).

### 2.2. Instruments

In this work, three data collection instruments were developed: 

Questionnaire assessing the level of importance, knowledge, concern, and action that young people attribute to their generation, past and their future offspring—with 16 closed-response questions on a 10-point Likert-type scale (1 = very low to 10 = very high), and one open question with the possibility of including comments.

Focus group interview guide (first stage)—in a semi-structured format, participants were invited to reflect on the work themes and the problems felt by their generation and future generations. 

Focus group interview guide (second stage)—similar to the previous one, the second stage focused on participant interviews promoting intergenerational debate and identifying strategies for the problems previously identified. Alongside, the questions were integrated: “How can communication and cooperation between generations be promoted?”; “What can young people do to have an impact on public policy by influencing decision-making processes related to their lives?”; “And because the COVID-19 pandemic had a great impact on the lives of all of us, how do you think this virus will affect the future generation?”.

## 3. Results

### 3.1. Data Analysis

The interventions of the focus group participants were registered by the co-interviewer and recorded by the participating schools. The audio of each focus group was sent by the participating school, facilitating the transcription process and making it as reliable as possible. The data were subsequently analyzed as qualitative data.

Quantitative data collected through a questionnaire was analyzed using SPSS v. 26. Based on a first content analysis, the rule of thumb principle was used (line-by-line procedure). A total of 1743 text segments from the focus groups were coded, leading to the categorization system of findings to define problems and risks (699 segments), strategies for intergenerational well-being (380 segments), and resources for greater intergenerational justice (222 segments). In addition, 33 text segments on engaging young people in political participation, 99 on promoting intergenerational communication and cooperation, and 104 on the impact of the pandemic on future generations were coded. 

### 3.2. Quantitative Study

Four areas were assessed regarding the issues under study: importance, knowledge, concern, and action. A total of 32 responses were obtained from the focus groups with young participants (mean age of 17.2 years, *SD* = 0.998, min. = 16 years, and max. = 19 years).

Overall (Table 1), young people attribute a higher regard for most issues under study (family and housing, work, sustainability, retirement, social security, education and health, and culture) to their generation when compared with older generations. The only topic on which they attribute a greater sense of importance to future generations is the environment.

In terms of knowledge, they self-assess themselves with a similar level of knowledge regarding the subjects of family and housing and a greater level of knowledge regarding culture when compared to older generations. In general, they believe that future generations will have a greater knowledge of most subjects (the only exception is culture).

When asked about their concerns, they are certain that their generation is the most concerned about all issues studied, with the exception of the environment.

In terms of action, the young people attribute to future generations a greater propensity for action on all the topics studied.

### 3.3. Qualitative Study

In the global qualitative study, based on the material collected in all the focus groups carried out in each of the five regions of mainland Portugal, the identification of numerous problems related to family and housing, education, environment, and work is observed. On the other hand, family and housing are also mentioned as important resources, especially in family relationships. A greater number of solutions were pointed out for the environment; young people seemed to more easily think of solutions for this topic. The topics of government and politics, as well as community and society, have fewer problems but also fewer resources and strategies gathered; they appear to be the themes that young people are more distant about.

A comparative study was also carried out with the goal of investigating the different perspectives of the 2002 and 2004 generations (voter and non-voter youth). Some of the problems, resources, and strategies identified were similar among young people from both groups.

In the topic of influencing public policies, young people were asked what they could do to influence public policies. Both groups (voters and non-voters) identified the school as a critical context for promoting youth political participation, specifically through the development of activities and political education. Both also mentioned the need for proactivity and participation of young people in contexts such as demonstrations, as well as the representation of young people in Parliament. Communication through social media was also mentioned by both groups as a possible means of expressing the needs of the younger generations.

Young voters referred to the exercise of the right to vote, the implementation by the government of means to listen to their generation (e.g., programs, platforms, or surveys), and the promotion of activities, not only at school but also in the community.

Regarding the identification of solutions for promoting intergenerational cooperation and communication, the young people of both groups pointed out open communication with adults (family members and in school context), and the creation of projects in schools.

Young voters also added the creation of activities and projects in the community, the use of technology and social media to bring the generations together, breaking prejudices, sharing knowledge, valuing others, education, a focus on common points between the generations, and the creation of a platform that promotes this communication.

As for non-voting youth, they mentioned the creation of opportunities for their generation to express their needs and a greater appreciation of their voice by adults.

At the closing of each session, and because the pandemic was and continues to be a reality with a strong impact on the lives of young people, they were asked about the impacts of COVID-19 on subsequent generations.

With similar comments on most of the identified consequences, both groups reported: (a) an increased sedentary lifestyle; (b) the impact on relationships (social and with family members); (c) an increased use of technologies; (d) the possible maintenance of teleworking; (e) the economic impact and economic crisis; (f) the maintenance of health care and sanitation; (g) the impact on health (both physical and mental); (h) the increasingly difficult access to the labor market. 

Young voters also mentioned some additional, both positive and negative, consequences: the impact on education and the fact that the focus on the pandemic has left other important issues (such as the environment and climate change) in the background, on the one hand; and the possibility of closer family relationships, the positive impact on the environment, and a greater focus on collective well-being, on the other hand.

Non-voters pointed to additional negative outcomes, such as increased and unnecessary consumerism, the negative impact on culture, the emergence of new diseases and future pandemics, and a decreased sense of freedom; but also, the immunity of future generations to the virus, on the other hand.

A comparative analysis of the problems of the two focus groups (first and second moment), before and after young people were asked to dialogue with people from other generations in their life contexts, was conducted. The following points were observed: a better preparation for the identification of strategies in the second moment; and greater awareness of the problems and resources of the previous generations in the second moment.

In the first moment, several times they attributed the “blame” of current problems to past generations, especially environment-related issues; young people’s interviews with people from other generations contributed to a rethinking of this perspective with contributions from the other generations’ perspectives. With this intergenerational dialogue with parents, grandparents, and other older people of reference, young people contextualized some issues differently (e.g., at the time there was no environmental awareness and also fewer resources).

Young people seemed to understand that attitudes that protected the planet were still adopted back then (e.g., the use of cloth bags to buy bread, glass bottles of milk, or the lesser use of own vehicles, which not all families could afford, for example). 

The family issue was also addressed—they understood that today they have parents with a more open mindset and that it is possible to talk to them and express their opinions; on the other hand, nowadays their parents spend less time at home and have less time for their children. They also discussed gender roles, arguing that if a few years ago, the man was the provider for the family, nowadays both parents have to work to meet the family expenses.

Access to housing was also discussed, making them reflect that at the time of their parents and grandparents it was more affordable. Regarding the work market, they compared the lifelong jobs of the past with today’s reality. They also considered themselves a more differentiated generation in terms of qualifications, and reflected on the fact that access to education is now easier and more generalized.

The topic of gender inequality was also discussed, but according to young people, past generations are not aware of this issue.

Regarding health, they are unanimous in their comments and the general idea that health care today is better and more accessible.

From the intergenerational interviews and debate of ideas, more sustained and organized strategies emerged at the second moment (second focus group). Young people are certain that they are better prepared and sensitive to deal with environmental issues than previous generations, and they believe that awareness should start at an early age. Even if they feel little support from adults regarding their concerns about the environment, they will continue spreading information and awareness to previous generations.

In the family, as strategies to prevent the sharp decline in birth rates over the years, they point to the need to provide more support to families, as well as the creation of more jobs.

They reiterate the importance of support from the government regarding the difficulty of accessing housing and the delay in achieving financial autonomy, attributed to the fact that they start to work later, and that employability is decreasing. Pressured by excessive evaluation and the choice of a professional area at an early age, they think that the educational system must be updated and reformulated since it has been maintained since the time of their grandparents. They believe that in addition to valuing learning, the school should promote life skills as well.

In health, they advocate for a restructuring and organization of primary health services and consider that mental health should be an investment target and free of charge. From the intergenerational debate, they believe that their parents and grandparents are not aware of the importance of this issue.

Aware of their grandparents’ reality in terms of access to retirement and pensions, they think that this will be a major problem for the following generations. They are concerned that they may become the first generation without access to retirement, partly because they will enter the world of work later, prolonging the need to work longer. They argue that access to retirement should be assessed on a case-by-case basis.

With the feeling that they are little or not at all heard by the bodies of political power, they support the integration of disciplines or activities of a political nature into the school curriculum, a strategy that they believe can increase the political participation of the current and subsequent generations.

Finally, regarding culture, and certain that this is one of the great national heritages, they support its modernization and access to it, since the prices charged are not always accessible to the entire population, and the role of schools in raising awareness of its importance. They also believe that culture can be an important vehicle in promoting intergenerational debate and that this is the way to ensure that national traditions do not become extinct.

## 4. Discussion and Conclusions

From a young person’s perspective…

Throughout the investigation and, in particular, in the early stages, it is clear that the younger generation self-assesses as the most worried and dedicated of all the groups involved regarding the main problems of our society (those approached with this methodology). Though it is not entirely logical from a practical standpoint, this generalized concern is significantly higher among young people, considering they are not directly affected by some of the problems under the lens in this paper (e.g., pensions, taxes, and housing). The fact that this generation claims to have a deeper understanding of said realities may reflect either increased interest and future planning concerns (which is a given at this point) or an exaggerated self-evaluation of their familiarity with said subjects (recall that one common complaint among teenagers is a lack of life skills training in school). According to Cassidy et al. [31], even though school standards and curriculum change over time, the need to promote life skills in students remains constant, a reality that the country still lacks.

They claim their descendants will have fewer concerns on every subject except for the environment, from which we can perhaps deduce that (i) they are confident of the changes to be introduced once they reach influential positions, and (ii) environmental issues’ awareness is meant to be a flag in future schools and communities. However, although aware of environmental issues [32], according to Wray-Lake et al. [33], young people tend to attribute the responsibility for the environment and planet to the government and consumers, not assuming their responsibilities. On the contrary, they consider that the following generations will value little culture, something that already happens in the country, and that this guarantee that can encourage greater intergenerational justice [34] will be lost over the years.

This generation naturally acknowledges their lack of direct influence and veto rights on political issues and change-making, claiming a small share of the parliament. In a study related to youth apathy in democratic life, it is suggested that young people are willing to engage but have no interest in the focus and nature of discourses as well as political life, which they believe ignore their needs and interests [35]. Their suggestions can be vague sometimes, though they emphasize the potential of education to aid them in developing, among other things, political literacy. The role of the school is recognized in promoting youth participation and political education [36]. This is supported by the fact that there is a common and generic appeal to schools, chambers/councils, and politicians to dynamize rarely further described activities to “solve” the issues they identify.

Let’s note that, as a result of our developing society, the current state of things is undeniably an improvement over the old status quo. Although it lacks changes, as is the case of the economy; housing, which is still a problem for the current generation, as a result of the scarce job opportunities; and education, which seems to be losing relevance, due to the excessive focus on assessment and perpetuation of an education model that is over 50 years old [36], today’s young people are characterized as more flexible, better educated, and more entrepreneurial, a profile that could be changed [37] for better or worse, based on the pandemic period currently being faced; health, a problem that essentially affects previous generations, seems to have benefited greatly, even more so with the technology implemented during the pandemic.

Concerning the concept of family, it is difficult to understand whether the change was positive or negative, and although the traditional family of the 21st century is more open and liberal, it is also less united than previous generations. Another study [38] confirmed this.

Regardless of this, teens tend to identify more issues regarding the topics under investigation, although, in reality, the problems one can raise about these topics are more scarce, which means the issues they identify are more specific, suggesting this generation is more critical of the world they live in.

When the debate is set between the new and the old generations, though, a true phenomenon occurs. After changing insights with their older generations, teens demonstrate a clearer understanding of the real problems they should be identifying and solving. There is now a true understanding of the intergenerational “crisis,” and the remarkable effect of the intergenerational debate truly highlights the importance of said debate—e.g., the importance of an exchange of ideas between teens and adults. Intergenerational dialogue, focused on a knowledge and action approach, has proven extremely relevant in improving resources and opportunities, thus supporting the achievement of greater intergenerational justice [39].

Despite the lack of actual decision power and sometimes less ambiguous suggestions, the current generation is not lackluster in willpower—they promise to keep on raising awareness on the issues that matter to them and they will strongly persist in facilitating connection and discussion with adults, making sure they have their insights considered in political and public affairs. This is no different between voters and non-voters. It is an ambitious generation that aims to prepare themselves and future generations to effectively deal with the past generations’ mistakes whilst assuring intergenerational justice. In the opinion of stakeholders linked to youth work, it is recognized that their participation in public policies should go through social marketing [36], thus supporting behavioral change and positively influencing society [40].

As a result of this reflection, which took place during an unprecedented period, it is now even more urgent to prioritize intergenerational dialogue and consequent intergenerational justice, as a way of supporting the minimization of the impacts of the pandemic on current and future generations.

## 5. Key Messages

Intergenerational justice has proved to be an important resource in the deconstruction of beliefs, in raising awareness of the problems of previous generations, and in increasing the knowledge that enables change.Young people believe in their knowledge and recognize the importance of the issues being discussed by their generation. They attribute a greater propensity for action to future generations.Family and housing stand out in most of the problems identified, but also as an important resource in the promotion of greater intergenerational well-being.The environment gathers a larger number of strategies, but their implementation seems to be considered the responsibility of future generations.Issues related to government and politics, community and society, and the economy do not raise much interest and knowledge about them is scarce. Awareness-raising, training, and support in the development of political and life skills seems to be a good educational strategy, which will pass through schools and schooling.Increased intergenerational justice will be achieved through the creation of means to facilitate and promote intergenerational dialogue, such as programs, platforms, or questionnaires.Aware of their low participation and few opportunities, young people believe that social networks could be a driving force in this action.Sensitive and alert to the impacts of the pandemic, young people advocate greater support for the psychological health and well-being of their generation.Holders of greater knowledge, ascribers of greater importance, and those more concerned with culture, believe that this will tend to be lost in the following generations, who, although better educated, will be less cultured.Voters and non-voters have similar and complementary discourses, showing that critical thinking is also a characteristic of the youngest generation.

## 6. Strengths and Limitations

As limitations of the study, we highlight the pandemic scenario in which it took place, forcing the development of an exclusively online methodology. In this context, the presence of one teacher per group was mandatory in order to have a facilitator; the issues of internet signal and sound sometimes reduced the time allocated to the groups; and the impossibility of control over the number of elements per group, as classes could not be separated by groups due to the containment measures imposed by the pandemic. However, the limitations are thought to be minimized by the study’s pertinence and innovation, which places young people in the first-person position in the identification of and action on current problems and those of future generations. In addition, the diversity and number of participants.

## Figures and Tables

**Table 1 children-09-01574-t001:** Means of importance, knowledge, concern, and action [30].

Topics	My Generation				Parents’ Generation				Grandparents’ Generation				Future Generations			
	*M*				*M*				*M*				*M*			
	I	K	C	A	I	K	C	A	I	K	C	A	I	K	C	A
Environment	9.2	7.6	8.5	7.1	6	5.4	5	4.8	4.5	3.8	3.6	3.5	**9.4**	**9.1**	**9.2**	**8.7**
Family and housing	**8.8**	**7.9**	**8.5**	7.5	7.7	7.3	7.5	7	7.3	6.4	6.6	6.5	8.1	**7.9**	8.2	**7.8**
Work	**8.5**	7.2	**8.6**	7.2	8.1	7.2	7.9	7.2	6.1	5.7	5.9	5.6	8.3	**7.9**	8.1	**8.1**
Sustainability, retirement, social security, education and health	**9**	7.5	**8.8**	7.1	8.2	7.5	7.8	7.8	7.2	6.3	6.9	6.3	8.6	**8.5**	8.2	**8.3**
Culture	**8.2**	**7.6**	**7.6**	6.8	7.5	6.8	6.7	6.5	6.8	6.2	6.5	6.2	7.7	7.5	7.1	**7.2**

Note: I = Importance; K = Knowledge; C = Concern; A = Action. The higher mean values of Importance, Knowledge, Concern, and Action in each topic are highlighted.

## Data Availability

Data supporting the reported results can be found here: https://gulbenkian.pt/de-hoje-para-amanha/o-que-pensam-os-jovens/, accessed on 17 October 2022.

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
