# Peer review of "We Are Concerned about the Future and We Are Here to Support the Change: Let’s Talk and Work Together!"

_children, 2022, doi:10.3390/children9101574_

Round 1
Reviewer 1 Report
Dear authors,
I truly appreciate the importance of the article about intergenerational justice, the empirical research is sound and original, the results are properly analyzed and conclusions are discussed against literature.
I am specifically grateful that the authors put interesting knowledge, literature review, empirical research and conclusions into 11 pages article. It is rather short, yet comprehensive. I do not have any particular comments, edits or suggestions.
Author Response
Dear Reviewer 1,
Thank you very much for your time in reviewing this work. The paper was reviewed.
Reviewer 2 Report
This paper aims to identify generational inequalities with the aim of promoting awareness of intergenerational challenges, boosting public debate and interest in public policies. It is a well-written paper.
The paper has some gaps that needs to be addressed before accepting it for publication
· Did this project received any kind of ethics approval?
· How the participants were recruited?
· Did the authors take an informed consent from the participants?
· When and for how long the data was collected?
· How the data was recoded and stored in an online FGD?
· Change heading to “Discussion and Conclusion”
· In the discussion part, compare your study results with the current literature.
· Mention limitations of the study
Author Response
Dear Reviwer 2,
Thank you very much for your time in reviewing this work, which I am sure has greatly enhanced this submission.
Enclosed are answers to the questions presented:
- Did this project received any kind of ethics approval?
Yes, it is mentioned on page 10, lines 490 and 491 “The Lisbon Academic Medicine Centre's Ethics Committee (35/20) approved this project.”
- How the participants were recruited?
Added - Lines 133 to 135 - The 15 participants were recruited through direct invitation via email, based on their experience and expertise.
Lines 143 to 147 - The participating schools were suggested by the Directorate-General for Education, and the participating classes were selected by the director of the educational institution, based on the availability of a class teacher to support the groups, and the age of the students (born in 2022 - 12th grade and in 2004 - 10th grade).
- Did the authors take an informed consent from the participants?
Lines 474 to 476 - This information was supplemented “Informed Consent Statement: Study participants or their guardians signed a free and informed consent form, which was passed on, collected and stored by the participating educational establishments.”
- When and for how long the data was collected?
Lines 134 to 135. Information added – “… in the period from May to October 2020.
Lines 146 to 147 – Information added – “The groups were conducted from November 2020 to January 2021.”
- How the data was recoded and stored in an online FGD?
Lines 188 to 191 – “The interventions of the group participants were registered by the co-interviewer and recorded by the participating schools. The audio of each group was sent by the participating school, facilitating the transcription process, and making it as reliable as possible. The data was subsequently analysed as qualitative data.”
- Change heading to “Discussion and Conclusion”
Changed.
- In the discussion part, compare your study results with the current literature.
References are scarce and such a comparison was attempted in the first version, but more literature was added in lines 360 to 362 and 374 to 377.
- Mention limitations of the study
Lines 453 to 463 – Section added.